# Long-Term Adverse Effects of Neck Radiotherapy in Childhood on the Carotid Arteries in Survivors of Hodgkin Lymphoma

**DOI:** 10.3390/cancers15153992

**Published:** 2023-08-06

**Authors:** Matjaž Popit, Marjan Zaletel, Bojana Žvan, Lorna Zadravec Zaletel

**Affiliations:** 1General Hospital Murska Sobota, Ulica dr. Vrbnjaka 6, 9000 Murska Sobota, Slovenia; matjaz54@gmail.com; 2Faculty of Medicine, University of Ljubljana, Vrazov trg 2, 1000 Ljubljana, Slovenia; marjan.zaletel@kclj.si (M.Z.); bojana.zvan@kclj.si (B.Ž.); 3Department of Vascular Neurology and Intensive Neurological Therapy, University Medical Centre Ljubljana, Zaloška Cesta 2, 1000 Ljubljana, Slovenia; 4Radiotherapy Department, Institute of Oncology Ljubljana, Zaloška Cesta 2, 1000 Ljubljana, Slovenia

**Keywords:** Hodgkin lymphoma, childhood, neck radiotherapy, carotid arteries, arterial stiffness, flow-mediated dilation

## Abstract

**Simple Summary:**

With novel treatments, the majority of Hodgkin lymphoma patients will become long-term survivors, which carries a risk for long-term sequelae due to treatment or the disease itself. We aimed to explore the late effects of therapeutic modalities on arterial stiffness and flow-mediated dilation in such patients, taking cardiovascular (CV) risk factors into consideration. In a group of 79 Hodgkin lymphoma survivors, we found increased arterial stiffness compared to healthy controls but not flow-mediated dilation. Neck radiotherapy increased arterial stiffness and anthracyclines decreased it. Our patient group also had more pronounced carotid atherosclerosis than controls. Our results show long-term vascular changes in Hodgkin lymphoma patients who might be therefore at increased risk of stroke. Systemic follow-up of these patients for carotid disease is warranted.

**Abstract:**

Introduction: Survivors of Hodgkin lymphoma are recognized to have an increased risk of stroke and carotid artery disease owing to neck irradiation (RT). However, it remains unclear whether the vascular modifications induced by the treatment of Hodgkin lymphoma during childhood persist over the long term. Methods: Our matched study involved 79 survivors of Hodgkin lymphoma in childhood who received neck RT and 57 healthy controls. Parameters of arterial stiffness (AS), intima-media thickness (IMT), and flow-mediated dilation (FMD) of carotid arteries were assessed using ultrasound. Results: Our patient cohort demonstrated a significant increase in AS compared to controls (*p* < 0.05), though no such disparity was observed for FMD (*p* = 0.111). Neck RT intensified AS (B = 0.037, *p* = 0.000), while anthracyclines attenuated it (B = −0.803, *p* = 0.000). Multivariate analysis revealed a positive correlation between neck RT (*p* < 0.001) and AS. However, we found no significant association between neck RT and FMD (*p* = 0.277). We identified a substantial positive correlation between the dose of neck RT and AS. Conclusions: Vascular changes in survivors of childhood Hodgkin lymphoma after neck RT seem to be long-term. Therefore, these patients may have an increased risk of stroke. We suggest refinement of international guidelines according to our results.

## 1. Introduction

The majority of Hodgkin lymphoma (HL) patients will become long-term survivors of their cancer with current therapies [1]. However, survivors of HL in childhood (HLSC) are at risk of several late effects of treatment, including secondary cancers and cardiovascular diseases [2,3,4]. 

Radiotherapy (RT) is an important treatment modality for several malignancies, including head and neck cancer [5]. The increased risk of carotid artery disease and stroke after RT in adult head and neck cancer and HD patients is well documented [6,7,8,9,10,11,12,13], but these radiation-related late effects have only recently been documented in adult survivors of childhood cancer [14,15,16,17]. Meeske and coauthors described a case series of young pediatric cancer survivors with advanced carotid stenosis after receiving neck RT in childhood [14]. The same author later found thicker carotid intima-media thickness (IMT) and a larger number of carotid plaques (CP) in pediatric cancer survivors than in controls [16]. Similar results were found in HLSCs vs. controls [15].

The main manifestation and adverse consequence of radiation-induced carotid injury (RICI) is carotid artery disease. The pathogenesis of atherosclerotic changes after irradiation is not entirely clear. Namely, most studies were carried out on animals or case series with a small number of patients. Some authors argue that chronic occlusive vasculopathy after RT is a consequence of accelerated atherosclerosis. They describe endothelial cell damage, fibrosis, intima thickening, scarring of the media, and fibrosis of the adventitia [18]. Others argue that this is the result of ischemia of vasa vasorum [19]. The main drivers of these processes seem to be arterial inflammatory injury, oxidative stress, epigenetic changes, and changes in surface protein expression [20]. While evaluation of RICI can be performed with neck auscultation, serum biomarkers, and imaging modalities, it is primarily accomplished through the use of ultrasound [17]. This diagnostic procedure is non-invasive, fast, relatively inexpensive, and has no radiation exposure [17]. While intima-media thickness (IMT) is most widely used for the evaluation of RICI, different ultrasound techniques such as arterial stiffness (AS) measurements or flow-mediated dilatation (FMD) were also described for this purpose [18,19]. Ultrasound can also be used for HLSC systematic follow-up. It is an effective, safe, and low-cost tool that is comparable to fluorodeoxyglucose (FDG) positron emission tomography (PET)/computed tomography (CT) regarding the detection of suspected malignant lymphadenopathy [20,21,22].

AS is the reduced ability of artery vasoconstriction and vasodilation due to changes in blood pressure. It has been regarded as a reliable marker of arterial structural and functional alteration after abundant experimental and clinical studies [21]. It is well documented that AS is an independent risk factor for cardiovascular disease and is associated with increased mortality because of cardiovascular disease [22], and it was shown to be associated with higher stroke incidence independently of cardiovascular factors, sex, and age [23]. Using ultrasound to measure AS in the common carotid artery seems to be a clinically applicable method concerning long-term follow-up studies [24]. Furthermore, the AS determined by the carotid ultrasound method has appeared as a possible surrogate marker for stroke in long-term survivors of childhood cancer [24].

Endothelial dysfunction is another important factor that increases the probability of stroke [24]. Endothelial impairment may be the first step of vascular toxicity and is considered the earliest step in the pathogenesis of atherosclerosis and thrombosis, which leads to cardiovascular diseases [25]. Endothelial dysfunction can be identified in all different clinical subtypes of stroke [26]. 

Recent findings on arterial stiffness in HL patients are not consistent. Van Leeuwen-Segarceanu and coauthors found an increase in pulse-wave velocity (PWV) and distensibility coefficient in HL patients treated with neck RT with a mean dose of 40 Gy [27]. In contrast, the study of Parr and coauthors suggested that in patients with lymphoma, AS improved with effective therapy [28]. A recognized method for assessing the endothelial function of peripheral arteries is flow-mediated dilation (FMD) [29]. The reports of some authors studying endothelial activity with FMD in cancer patients were contradictory as well [17,30,31]. Beckman et al. found a significant reduction in FMD in the axillary arteries of women who were irradiated to the breast and axilla for breast cancer [30]. Dengel et al. found a significant difference in brachial FMD between childhood cancer survivors treated for acute lymphoblastic leukemia with a combination of chemotherapy and cranial irradiation in comparison with those treated with chemotherapy only [31]. Brouwer et al. investigated vascular changes and brachial FMD in 277 childhood cancer survivors, of whom 174 (63%) received different types of RT (RT to the neck, chest, or mediastinum only; cranial RT; other) and 221 (80%) received chemotherapy. There were no differences in brachial FMD between childhood cancer survivors who were treated with radiation or chemotherapy and their healthy closest relatives [17]. There were also no differences in FMD between different treatment groups [17]. However, they did find an association between both carotid and femoral IMT regarding RT. A shortcoming of these studies was that they did not take into account traditional cardiovascular (CV) risk factors’ effects on AS. Our goal in the present study was to explore the late effects of therapeutic modalities on AS and FMD in HLSCs, considering CV risk factors.

## 2. Materials and Methods

Patients were eligible for our observational population case-control study (clinical trial number IP-0302) if they had been treated for HL in Slovenia between 1970 and 2005, at the age of 17 years or less, and had received RT to the neck. One hundred seventy-three patients were treated for HL at this age, thirty-five died, and 112 out of 138 living patients received neck RT and were eligible for the study. Nineteen patients live outside Slovenia and/or are not followed up at our outpatient department for long-term follow-up at the Institute of Oncology Ljubljana; another 14 patients refused to participate in the study. Information about diagnosis and treatment was abstracted from the patient’s medical records. The flow chart of patient recruitment is summarized in Figure 1.

We included 57 healthy controls (36 females, 21 males) who were not treated for cancer. They were between 20 and 61 (average 41.0) years old at the time of study. They were recruited through our neurology outpatient clinic, and we included those who were without significant neurological deficits and in a similar age range to the HLSCs.

Exclusion criteria for patients and controls were heart arrhythmia, signs of angina pectoris, recent infection, ongoing cancer, therapy with steroids, and anti-inflammatory drugs, as these factors could significantly alter our measurements [32]. No HLSC from our cohort met the exclusion criteria, though.

In our study, duplex Doppler sonography of the cervical arteries was performed in all subjects at the ALOKA α10 (Tokyo, Japan) with an 8 MHz probe. IMT was measured according to the Mannheim criteria [33]. The velocities and diameters of the common carotid arteries were measured 2 cm below the bulb or at the best visible site proximal to the bulb. We determined the presence of CP and its quality in both common carotid arteries. All measurements were made by the same examiner on the same ultrasound machine.

This study was performed in line with the principles of the Declaration of Helsinki. Approval was granted by the Ethics Committee of the Republic of Slovenia on 12 June 2018, and the number of inquiries was 0120-277/2018/5. Informed consent was obtained from all individual participants included in the study.

### 2.1. Traditional Cerebrovascular Disease Risk Factors

Of the traditional cerebrovascular disease risk factors, we asked the subjects about the presence of arterial hypertension (AH), diabetes mellitus (DM), hyperlipidemia (HPL), smoking, family history of stroke or myocardial infarction, height, and weight. A positive family history of cerebrovascular disease was defined as a cardiac or cerebral ischemic event in a first-degree relative younger than 65. Blood was drawn from the cubital vein for laboratory tests (4-fractional lipidogram, glucose, CRP, and fibrinogen). Blood samples were taken between 7 a.m. and 9 a.m. after a minimum of 12 h of empty stomach and smoking, alcohol, and caffeine abstinence. HPL was present if the subject had serum LDL cholesterol above 2.6 mmol/L and/or triglycerides above 1.7 mmol/L.

### 2.2. Arterial Stiffness (AS) Parameters

AS parameters were automatically measured on the same abovementioned US machine by modifying the arterial diameter between the systolic and diastolic phases on standard carotid artery segments. Carotid diameter waveforms were assessed using ultrasound and converted to carotid pressure waveforms using an empirically derived exponential relationship between pressure and arterial cross-section. Blood pressure measurements were obtained simultaneously with ultrasound measurements. The derived carotid pressure waveform was calibrated from brachial end-diastolic and mean arterial pressures by iteratively changing the wall rigidity coefficient. This allowed the calculation of the AS parameters obtained as mean values of the last six measurements. For the analysis of AS parameters, we used the same formulas as outlined in the previous article from our group [24].

### 2.3. Endothelium-Dependent Flow-Mediated Vasodilation

In all subjects, hemodynamic measurements (ALOKA Alpha 10, Tokyo, Japan) on the brachial artery were performed in a supine position after a minimum of 15 min, resting in a quiet place at a temperature of 22–26 °C. Before starting the investigation, we measured arterial blood pressure with a sphygmomanometer. During the investigation, we continuously recorded the ECG and performed all the measurements at the end of the diastole, at the R wave in the ECG. All measurements were made on the right brachial artery. The brachial artery was assessed transversely and then longitudinally using a 10 MHz linear probe. When the image was the sharpest with clearly visible front and back edges of the intima and vascular lumen, it was frozen, and the baseline diameter was measured. Then, we measured the average blood flow velocity with a pulse Doppler in the middle of the artery. Then, the forearm was spun with a sphygmomanometer cuff with a pressure of 250 mmHg for 4 min. The flow rate was measured within 15 to 20 s, and the dilated artery diameter was measured 60 and 90 s after the release of the cuff. All measurements in individual subjects were performed at the same time of the day. The same investigator carried out all measurements.

Endothelium-dependent FMD was calculated following the equation:$$FMD=\frac{Dpov-Dmir}{Dmir}$$

Dpov is the artery’s diameter following an increased flow, and Dmir is the arterial diameter at rest.

### 2.4. Statistical Methods

For statistical processing, the SPSS Statistics 26 program was used. The sample was determined by our previous pilot study (24). We used the chi-square test to test the differences in common CV risk factors. Differences between the two groups regarding carotid stiffness parameters, FMD, and CP were tested using the Student’s *t*-test for independent samples. The effect of chemotherapy was sorted out with multivariate analysis. We used linear regression to test the correlation between FMD and treatment modalities. To test the effect of the dose of neck RT on AS, we analyzed the PWW (as the representer of AS) and RT dose of each irradiated side of the neck with the linear regression method. Namely, 15 patients had only unilateral RT, and 6 patients received different doses to the left and right sides of the neck. Statistical significance was set at *p* < 0.05.

## 3. Results

Seventy-nine patients were investigated. Table 1 provides the detailed characteristics of the cohort. They were 3 to 17 (average 11.2) years old at diagnosis and had evaluation 14 to 47 (average 30.8) years later. The average age of participants in the control group was not significantly different from that of our HLSC group (*p* < 0.05). Groups were also well matched regarding hyperlipidemia (HPL), smoking, and family history of CV diseases. Differences between groups regarding common CV risk factors are summarized in Table 2.

The total dose that HLSCs received to the neck was between 10 and 60 Gy (median 30 Gy) in 1.5 to 2 Gy daily fractions. Three patients received RT with doses higher than 42 Gy (59–60 Gy) because they had relapsed and were irradiated to the neck twice. Sixty-five subjects received chemotherapy, 42 patients were treated with anthracyclines containing chemotherapy (ACC), and others received MOPP (14 patients), LOPP (7 patients), and COPP (3 patients). 

Our analysis showed a significant increase in carotid stiffness. The HLSC group also had more carotid plaques and thicker IMT. There was no difference in FMD between the two groups. Results are summarized in Table 3.

Linear regression showed a significant positive correlation between the dose of neck RT and PWW as a representer of AS (B = 0.043, R^2^ = 0.147, *p* < 0.001). The mean neck dose was 25.5 ± 12.4 Gy and PWW was 6.4 ± 1.4 m/s. 

At the end of our research, we tested multivariate relationships between treatment modalities and AS. We considered PWV as a representative marker for AS. In our first step, we tested the effect of RT and AH on PWV to identify any confounding effect. In our model, treatment with ACC and neck RT appeared significant but not AH (Table 4).

In our second model (Table 5), FMD was the dependent variable, while treatment with either ACC or neck RT was the independent variable. There was no significant connection between FMD, ACC, and neck RT (*p* > 0.05).

## 4. Discussion

In our present study, we evaluated the AS of carotid arteries in a group of HLSCs after neck RT and in a group of healthy controls. We found that AS was increased in HLSCs compared to controls. 

We found a positive correlation between neck RT and AS in line with our expectations and reports in the literature [24,27,28,34,35,36]. A systematic review [37] demonstrated an increased incidence of stroke/TIA in patients receiving neck RT for HL. Furthermore, the same meta-analysis showed consistent differences in carotid AS and IMT between irradiated and unirradiated carotid arteries. The main issue was that the majority of studies utilized sub-optimally matched controls for each endpoint [7,37,38,39,40]. We also found a significant positive correlation between neck RT dose and AS which has not been described so far. This finding calls for further studies confirming these results, as modern RT treatment techniques deliver smaller doses to more precise locations (involved nodal RT).

Anthracyclines are very effective chemotherapeutic agents in the treatment of HL, and ACC is usually applied as frontline treatment. Anthracyclines have many acute and long-term side effects, cardiovascular side effects being of most concern [41]. These are thought to arise due to cardiomyocyte death through free radical formation caused by anthracycline metabolism [41]. Similarly, anthracyclines can alter AS, as Herceg-Cavrak et al. reported an increase in aortic PWV in childhood cancer survivors treated with anthracyclines compared with healthy controls. However, the average follow-up time in this study was 2 years only, and AS was measured with PWV only [42]. Moreover, the risk of stroke is also increased after treatment with anthracyclines due to endothelial dysfunction induction and AS increase [43]. We measured carotid PWV rather than aortic, which could be a marker for systemic arterial stiffness.

This brings us to our results which point to the unexpected finding of ACC being a possible protective factor against AS. Since our subjects were followed up on average 30 years after treatment, our group suggests two feasible reasons for this. Vascular endothelium repair does occur after vascular injury. This has been extensively researched after surgical or traumatic vascular injury [44,45] but not after radio- or chemotherapy. While ACC-induced vascular injury is well documented, it is not known if HL itself can damage the endothelium as well. Indeed, HL is a unique hematopoietic neoplasm characterized by cancerous Reed–Sternberg cells in an inflammatory background. It is also suspected that it causes systemic inflammation that could affect various organ systems, including arteries. We do not have data on the natural course of HL and its influence on AS without oncologic intervention, including chemotherapy. Nevertheless, it seems the combined effect of HL and cancer treatment, including neck RT on the arterial wall, could result in increased AS.

Further analysis showed that treatment with RT increased AS and that ACC decreased AS, but neither of the treatment modalities had a significant effect on endothelial dysfunction. Indeed, FMD is measured in the brachial, while AS is measured in the carotid artery. It is well known that AS is not uniform across all arteries [46]. This might explain the difference between our findings in the brachial and the carotid artery. Nevertheless, the endothelial function using FMD was assessed on average more than 30 years after diagnosis. Since endothelium is a highly viable tissue, its function could be restituted ad integrum. This might be a reason for no correlation between FMD and ACC. In addition, this could be the reason why resolved HL does not have a permanent effect on FMD, as we did not find significant differences in FMD between HLSCs and controls.

Treatment of HL patients has changed significantly in recent years in the direction of decreasing the RT field size and reducing the dose of RT. Our patients were treated before 2006, when the concept of involved-node RT for early-stage Hodgkin lymphoma was introduced [47,48]. We can expect that this will further reduce the adverse effect of RT on the carotid arteries.

In the newest European guidelines for follow-up of childhood and adolescent cancer survivors, there is no advice regarding follow-up of changes on carotid arteries after neck RT (48). According to the findings of the present study, we would suggest including US of neck arteries in the regular follow-up of HLSCs who received neck RT, taking into consideration presence of cardiovascular risk factors as well. 

According to the Guidelines on the Management of Patients with Extracranial Carotid and Vertebral Artery Disease [49], US of neck arteries might be considered to detect carotid stenosis in asymptomatic patients without clinical evidence of atherosclerosis who have ≥2 of the following risk factors: AH, HPL, tobacco smoking, family history in a first-degree relative of atherosclerosis manifested before age 60 years, or family history of ischemic stroke. We suggest adding neck RT to this list as an important risk factor for carotid stenosis. 

The main strength of our study is that it is a population-based study with a long follow-up. The main limitation could be a smaller control group.

## 5. Conclusions

Our study shows additional evidence of local adverse effects of neck radiotherapy on the carotid arteries in HLSCs even 30 years after treatment. This presents a significant burden of the disease and its treatment regarding long-term comorbidities such as stroke or cardiovascular events. Our findings support regular long-term follow-up for these patients, caring for a healthy lifestyle, regular monitoring, and reduction of cardiovascular risk factors.

We suggest adding neck RT as another risk factor for carotid stenosis into the Guidelines on the Management of Patients with Extracranial Carotid and Vertebral Artery Disease and to introduce US of neck arteries into European guidelines for follow-up of childhood cancer survivors who received neck RT. 

## Figures and Tables

**Figure 1 cancers-15-03992-f001:**
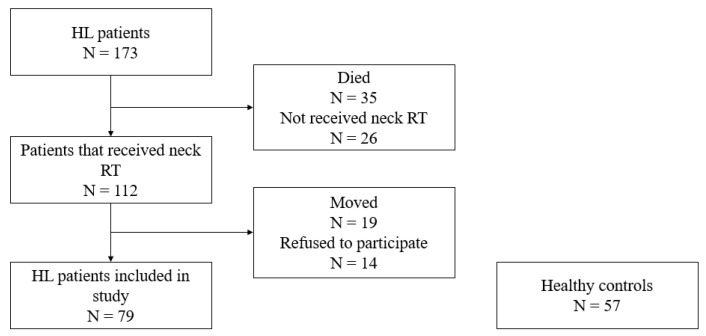
Flowchart of patient inclusion.

**Table 1 cancers-15-03992-t001:** Patients’ characteristics. HL: Hodgkin lymphoma, ACC: anthracyclines containing chemotherapy, RT: radiotherapy.

	Patients (%)
Sex	
Female	31 (39.2)
Male	48 (60.8)
Age at diagnosis of HL (years)	3 to 16 (average 11.2)
Age at evaluation (years)	22 to 64 (average 41.8)
Follow-up time (years)	14 to 47 (average 30.8)
Ann-Arbor stage	
I	17 (21.5)
II	40 (50.6)
III	19 (24.1)
IV	3 (3.8)
B symptoms	12 (15.2)
Bulky disease	19 (24.1)
Relapse	7 (8.9)
Treatment	
Chemotherapy	65 (82.3)
ACC	42 (53.1)
Prescribed RT dose to the neck (Gy)	10 to 60 (median 30)
staging laparotomy with splenectomy	20 (25.3)

**Table 2 cancers-15-03992-t002:** Differences between groups regarding common cerebrovascular risk factors. AH: arterial hypertension, DM: diabetes mellitus, HPL: hyperlipidemia.

		Patients (%)	Control Group (%)	Chi-Square	*p*
Sex	female	31 (39.2)	36 (63.2)	14.495	<0.001
	Male	48 (60.8)	21 (36.8)		
AH		23 (29.1)	1 (1.8)	16.927	<0.001
DM		4 (5.0)	0	4.026	0.040
HPL		63 (79.7)	43 (75.4)	0.693	0.405
Smoking		27 (34.2)	23 (40.4)	0.554	0.457
Family history		14 (17.7)	12 (21.1)	0.714	0.398

**Table 3 cancers-15-03992-t003:** Student’s *t*-test for independent samples for testing differences between groups regarding stiffness parameters, FMD, and age at evaluation. Beta: beta stiffness index, Ep: elasticity module, AI: augmentation index, AC: arterial compliance, PWV: pulse-wave velocity, FMD: flow-mediated dilation, CP: carotid plaques, IMT: intima-media thickness, SD: standard deviation, SE: standard error.

	HLSC (Yes/No)	N	Mean	SD	SE	*t*	*p*
Beta	yes	79	8.408	3.5077	0.3947	2.849	0.005
	No	57	6.911	2.1769	0.2883		
Ep	yes	79	119.447	58.7368	6.6084	3.360	0.001
	No	57	91.263	27.7499	3.6756		
AI	yes	79	10.524	12.5953	1.4171	1.772	0.079
	No	57	6.675	12.3515	1.6360		
AC	yes	79	0.6576	0.28555	0.03213	0.738	0.462
	No	57	0.6912	0.22597	0.0299		
PWV	yes	79	6.448	1.5187	0.1709	3.312	0.001
	No	57	5.714	0.8249	0.1093		
FMD	yes	79	4.4803	2.28793	0.25906	1.605	0.111
	No	57	5.1496	2.53294	0.33550		
CP	yes	79	0.899	1.5241	0.1715	3.809	<0.001
	No	57	0.105	0.4506	0.0597		
IMT	yes	79	0.895	0.2287	0.0257	2.115	0.036
	No	57	0.818	0.1824	0.0242		
Age	yes	79	41.788	9.0857	1.0158	−0.528	0.598
	No	57	40.982	8.3548	1.1066		

**Table 4 cancers-15-03992-t004:** Multivariate analysis of PWV and treatment options with testing AH as a possible confounder. ACC: anthracyclines containing chemotherapy, RT: radiotherapy, AH: arterial hypertension.

Model	Unstandardized Coefficients		Standardized Coefficients	T	Sig.
	B	Std. Error	Beta		
ACC	−0.803	0.215	−0.307	−3.736	0.000
Neck RT	0.037	0.007	0.481	5.427	0.000
AH	0.444	0.279	0.135	1.592	0.114

**Table 5 cancers-15-03992-t005:** Linear regression models show a correlation between FMD and different treatment modalities. ACC: anthracyclines containing chemotherapy, RT: radiotherapy.

Model	Unstandardized Coefficients		Standardized Coefficients	*t*	Sig.
	B	Std. Error	Beta		
ACC	1.231	0.727	0.145	1.693	0.093
Neck RT	0.024	0.022	0.095	1.092	0.277

## Data Availability

The datasets generated during and/or analyzed during the current study are available from the corresponding author upon reasonable request.

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
