# Peer review of "Long-Term Adverse Effects of Neck Radiotherapy in Childhood on the Carotid Arteries in Survivors of Hodgkin Lymphoma"

_cancers, 2023, doi:10.3390/cancers15153992_

Round 1

Reviewer 1 Report (Previous Reviewer 2)

If this is a case-contro study, is there a prospective arm?

Why does the prospective arm include 79 patients (sample size calculation, according the primary end-point of the study)?

What is the period of study for the prospective arm (between 1970-2005?)

Whai is the clinical.trial number?

What is the Institutional Review Board Statement for the study or Ethical Committee (number of acceptance) ?

Was informed consent obtained from all subjects? 

References should be improved

If this is a case-contro study, is there a prospective arm?

Why does the prospective arm include 79 patients (sample size calculation, according the primary end-point of the study)?

What is the period of study for the prospective arm (between 1970-2005?)

Whai is the clinical.trial number?

What is the Institutional Review Board Statement for the study or Ethical Committee (number of acceptance) ?

Was informed consent obtained from all subjects? 

References should be improved

Author Response

Response to Reviewer 1

Point 1: If this is a case-control study, is there a prospective arm?

Response 1: This was an observational case-control study, so there was no prospective arm.

Point 2: Why does the prospective arm include 79 patients (sample size calculation, according the primary end-point of the study)?

Response 2: Our study was a population based study. We included every HL patient that was treated in that time period and who agreed to participate and signed informed consent.

Point 3: What is the period of study for the prospective arm (between 1970-2005?)

Response 3: Period of the study was between 2019 and 2023. The period you are mentioning is the period when the patients were diagnosed with HL.

Point 4: Whai is the clinical.trial number?

Response 4: The clinical trial number is now stated in the Methods section.

Point 5: What is the Institutional Review Board Statement for the study or Ethical Committee (number of acceptance) ?

Response 5: Approval was granted by the Ethics Committee of the Republic of Slovenia on 12th June 2018, and the number of inquiries was 0120-277/2018/5. This is now stated in the text.

Point 6: Was informed consent obtained from all subjects?

Response 6: The informed consent were obtained from all participants. In fact, this was obligatory for participation in our study.

Point 7: References should be improved

Response 7: We accepted your suggestion and updated the references.

Reviewer 2 Report (Previous Reviewer 1)

Dear authors

thanks a lot for the corrections you made. The article has been well improved.

I still have a few remarks

method lines 132-3"Exclusion criteria for patients and controls were heart arrhythmia, signs of angina pectoris, recent infection, ongoing cancer, therapy with steroids, and anti-inflammatory  drugs as these factors could significantly alter our measurements." Could you add citation for this ?

In the method part, could you also detail what cardiovascular risks you took into account and how it was measured/collected as in the introduction you presented your work as new because of this "Our goal in the present study was to explore the late effects of therapeutic modalities on AS and FMD in HLSC considering CV risk factors."

Also in the method part, it is not well precised if you compare two groups of patients or if it is a matched study. 

Results : table 1 for RT is it prescribed dose ? or estimated dose received by the arteries ? Was rt bilateral ? did you consider this ? and what about the size of the field ?

Lines 205-206 should be in the method / statistical part

Multivariate analysis:

There is no result concerning the dose of radiotherapy. The risk did not increase with the dose ? 

There is no mention of the CV risk factors - as it was suggested in the introduction. This should be present or the introduction should me modified

Discussion

Line 259 you add "net". I am not sure what you wanted to emphasize with this.

Conclusion

"Our findings support regular long-term follow-up for these pa- 282 tients, caring for a healthy lifestyle and reducing cardiovascular risk factors."

The article is very interesting but I was a bit diasppointed of the conclusion which does not give new information - and just remind what we know.

What about performing ultrasonography ? Could you give recommendation on this point ? Way to perform ? Frequency ? Things that the radiologist should be looking at...

Author Response

Response to Reviewer 2

Point 1: method lines 132-3"Exclusion criteria for patients and controls were heart arrhythmia, signs of angina pectoris, recent infection, ongoing cancer, therapy with steroids, and anti-inflammatory  drugs as these factors could significantly alter our measurements." Could you add citation for this ?

Response 1: We added the reference according to reviewer suggestion.

Point 2: In the method part, could you also detail what cardiovascular risks you took into account and how it was measured/collected as in the introduction you presented your work as new because of this "Our goal in the present study was to explore the late effects of therapeutic modalities on AS and FMD in HLSC considering CV risk factors."

Response 2: we agree with the suggestion and described cardiovascular/cerebrovascular risk factors in the Methods section.

Point 3: Also in the method part, it is not well precised if you compare two groups of patients or if it is a matched study.  

Response 3: We additionally precised the method part. Basically, our study was a matched study.

Point 4: Results : table 1 for RT is it prescribed dose ? or estimated dose received by the arteries ? Was rt bilateral ? did you consider this ? and what about the size of the field ?

Response 4: It was prescribed dose and is now noted in table 1. We didn’t estimate the dose to the arteries, but the whole length of the neck was irradiated in our patients because this was just before the introduction of involved nodal RT. We stated in the methods now how many patients had unilateral neck RT and yes, we consider this in data manipulation, because we measured US parameters separately for left and right side of the neck.

Point 5: Lines 205-206 should be in the method / statistical part.

Response 5: Suggestion accepted, those lines were moved to Methods section.

Point 6: There is no result concerning the dose of radiotherapy. The risk did not increase with the dose ? 

Response 6: We accepted your suggestion and performed additional statistics which shows significant relationship between increasing RT dose and PWV as representer of AS.

Point 7: There is no mention of the CV risk factors - as it was suggested in the introduction. This should be present or the introduction should be modified.

Response 7: This was a matched study, tables 2 (CV risk factors other than age) and table 3 (age) show how well the groups were matched regarding CV factors. In the multivariate analysis arterial hypertension (AH) was chosen as the possible confounder since it was the least well matched risk factor.

Point 8: Discussion

Line 259 you add "net". I am not sure what you wanted to emphasize with this.

Response 8: We meant »combined«. It is now changed in the text.

Point 9: Conclusion

"Our findings support regular long-term follow-up for these pa- 282 tients, caring for a healthy lifestyle and reducing cardiovascular risk factors."

The article is very interesting but I was a bit disappointed of the conclusion which does not give new information - and just remind what we know.

What about performing ultrasonography? Could you give recommendation on this point? Way to perform? Frequency? Things that the radiologist should be looking at...

Response 9: The suggestion is reasonable, therefore we added what the joint guidelines from multiple ultrasonography societies suggest and added our recommendation. Furthermore, we added our recommendation to introduce the section about follow-up of survivors with US of neck after neck RT into European guidelines for long-term follow-up of childhood and adolescent cancer survivors.

Reviewer 3 Report (Previous Reviewer 3)

The manuscript has been improved. I suggest to shortening the introduction section and transferring some of the information to discussion section.

Quality of English Language is well.

Author Response

Response to Reviewer 3

Point 1: The manuscript has been improved. I suggest to shortening the introduction section and transferring some of the information to discussion section

Response 1: As suggested we shortened the introduction section and put some information in discussion.

Round 2

Reviewer 1 Report (Previous Reviewer 2)

Accept in present form

This manuscript is a resubmission of an earlier submission. The following is a list of the peer review reports and author responses from that submission.

Round 1

Reviewer 1 Report

Dear authors,

Thanks a lot for this study and this paper with important concerns.

Please find here a few comments to improve your article. In general, I suggest to review all the abreviations as they are not all explained, and be more precise.

Introduction:

This part of the article could be really improved, precising more data about CAYAC survivors, less about adults and more details about radiotherapy treatment. In addition, I suggest to tell the readers what is new in this analysis in comparison with the one you did in 2018. below a few suggestions with more details.

I suggest in the first sentence to write at the present and not "will" as your second sentence is on "old" data.

I suggest also to keep in the background about HL and Childhood or AYA curvivors as we can find some data in the litterature, and delete the reference 4 and 18 and change the sentences that go with. 

Or you could do as you did in 2018 and describe more the references of the second part of the sentence

"The increased risk of carotid artery disease and stroke after radiation therapy (RT) in adult head and neck cancer and HD patients is well documented7,8,9,10,11,12, but these radiation-related late effects have only recently been documented in adult survivors of childhood cancer.13,14,15,16"

Please give more details about the reference 17 (population, dose prescribed...)

line 89, please explain FMD before use abreviations (explanation is line 163)

material, method:

Lines 114, 115 should be in the result part

Control group should be mode detailed : way of recruitment... how does this ensure that this group is a control group of a langda population...

about exclusion criteria : why ? for who ? patients and controls ? how many patients were excluded because of these criteria ?

line 123 please explain RICI

lines 123-128 should be in the background, not really in method

line 133 CP is not obvious initials

statistical method is a bit light. Could you precise more

results

line 176 - match ? really ? it was not in the method. Do you mean no difference ?

table 1  Family history of what ?

HLSC is ?

line 181 "All HLSC received RT to the neck." it is evident as it is in the inclusion criteria. I suggest to delete

Line 181: until 65 Gy ?? could you precise how many with such doses and why ? About dose, it it only at first line treatement or sometimes because of relapse ? Did some patients had several RT treatment ? If yes how did you manage ?

table 2 please explain SD, SE, 

please reformulate  lines 195-197 and put them in the method part. Neck RT is not feasable as you only included patients with neckRT. It is not possible to do mutltivariate analysis including patients from a control group.

Ragrding your past study published in 2018 with patients not treated with modern radiotherapy and here with patients treated before 2000 and after, analysis should be more detailed.

Reviewer 2 Report

- Extensive editing of the english language is required;

- There are too many acronymus across the manuscript. These should be removed, and anyway explained at the first;

-In the Introduction section, the role of ultrasonography and power Doppler ultrasonography at baseline and during follow-up of patients with Hodgkin lymphoma should be emphasized. Please add in the bibliography the following references: Picardi M et al     Journal of Clinical Oncology 2004; 22: 3733-3740 / Picardi M et al Haematologica 2006; 91: 960-963 / Picardi M et al Radiology 2014; 272: 262-274;

- In Materials and Methods section and across the study, the authors should clearly specified the type of the study: retrospective, case-control or prospective trial. In these last two cases, the authors should provide the clinical trial number and the sample size assessment in the statistical sub-paragraph; 

- In the Results section, Figura 1 with flow-chart of the patients across the study should be added, as well as Table 1 with patients and disease characteristics, i.e. HL sub-types, Ann arbor stage, bulky disease, B symptoms, IPS, type of front-line chemotherapy regimen, and so on;

- Intra-observer and interequipment variability of color-Doppler ultrasonography results should be discussed, as well as the role of modern radiotherapy such as involved nodal irradiation.

- Extensive editing of the english language is required;

Reviewer 3 Report

Authors report long-term adverse effects of neck radiotherapy in childhood on the carotid arteries in survivors of Hodgkin lymphoma. The manuscript is potentially interesting, but requires a minor revision.

1. I suggest moving the last paragraph (line 86-103) of the section introduction to the discussion section.

2. I suggest to revise and more actually references.

1. The manuscript should be edited for proper English language, grammar, punctuation, spelling, and overall style. For example:

Abstract: Line 27: “– Methods”, please remove “-“;

Introduction: line 122: Duplex Doppler sonography of the carotid arteries…..??